# Molecular Mimicry Between *Toxoplasma gondii* B-Cell Epitopes and Human Antigens Related to Schizophrenia: An In Silico Approach

**DOI:** 10.3390/ijms262110321

**Published:** 2025-10-23

**Authors:** Juan F. Cano, Maria Andrea Bernal-Valencia, Pablo Vargas-Acevedo, Germán Mejía-Salgado, Andrés Sánchez, Oscar Correa-Jiménez, Marlon Múnera, Alejandra de-la-Torre

**Affiliations:** 1UR Center for Mental Health (CeRSaMe), School of Medicine and Health Sciences, Universidad del Rosario, Bogotá 111221, Colombia; juanf.cano@urosario.edu.co; 2Ophthalmology Interest Group Universidad del Rosario (OIG UR), School of Medicine and Health Sciences, Universidad del Rosario, Bogotá 111221, Colombia; mariaand.bernal@urosario.edu.co; 3Facultad de Medicina, Pontificia Universidad Javeriana, Bogotá 110311, Colombia; pablovargas@javeriana.edu.co; 4Neuroscience Research Group (NEUROS), Neurovitae Center for Neuroscience, Institute of Translational Medicine (IMT), School of Medicine and Health Sciences, Universidad del Rosario, Bogotá 111221, Colombia; germanmejiainvest@gmail.com; 5Medical Research Group (GINUMED), Universitary Corporation Rafael Nuñez, Cartagena 130016, Colombia; andres.sanchez@curnvirtual.edu.co (A.S.); marmunera@gmail.com (M.M.); 6Pulmonology and Immunology in Pediatrics Research Group, Department of Pediatrics, Universidad Nacional de Colombia, Bogotá 111321, Colombia; olcorreaj@unal.edu.co

**Keywords:** psychotic disorders, parasites, epitopes

## Abstract

Schizophrenia is a complex disorder influenced by genetic, neurobiological, and environmental factors, with increasing evidence implicating immune dysregulation. This study examined potential molecular mimicry between autoantigens associated with schizophrenia and proteins from *Toxoplasma gondii*, a parasite previously linked to the disorder. Amino acid sequences of schizophrenia-related autoantigens were retrieved from databases (AAgAtlas, PubMed), and homologous sequences were searched within the *T. gondii* proteome. Sequence identity was evaluated, and conserved B-cell epitopes were predicted using three-dimensional structures from the Protein Data Bank or models generated in Swiss-Model, followed by epitope mapping with ElliPro. Five autoantigens—gamma-enolase (*ENO2*), thyroid peroxidase (*TPO*), glutamic acid decarboxylase 65 kDa isoform (*GAD65*), serine/threonine-protein kinase 2 (*VRK2*), and dihydropyrimidine dehydrogenase [NADP(+)] (*DPYD*)—showed similarities with *T. gondii* proteins. Among them, enolase exhibited the highest homology, with identities up to 65%. These findings provide preliminary evidence of shared antigenic features between the parasite and schizophrenia-related autoantigens. Such mimicry could contribute to disease mechanisms by triggering autoimmune responses in genetically susceptible individuals, supporting the hypothesis that *T. gondii* infection may influence schizophrenia pathogenesis. Nonetheless, the results are based exclusively on in silico analyses, and experimental validation will be required to confirm potential cross-reactivity.

## 1. Introduction

Schizophrenia is a complex and debilitating neuropsychiatric disorder affecting approximately 1% of the global population [1]. It is influenced by genetic, neurobiological, and environmental factors [2]. While genetic predisposition, neurochemical imbalances, and environmental stressors such as prenatal stress or toxin exposure are known contributors, the precise underlying causes remain elusive due to the multifaceted interaction among these factors [2,3,4]. Immune dysregulation has been implicated in the pathogenesis of psychiatric disorders, including schizophrenia [5]. Key mechanistic pathways implicated include microglial activation, elevation of pro-inflammatory cytokines, molecular mimicry, anti-neuronal autoantibodies, self-reactive T-cell responses, and disturbances in blood–brain barrier integrity [6,7].

Infectious agents, such as *Toxoplasma gondii*, are significant environmental factors that increase the risk of developing schizophrenia [8,9]. Epidemiological and experimental data support this association. Meta-analyses have shown that individuals with schizophrenia are more likely to be seropositive for *T. gondii* compared to the general population, with odds ratios of 2.73 (95% CI 2.10–3.60) [10]. Furthermore, animal models demonstrate that *T. gondii*-infected mice exhibit behavioral changes, such as increased attraction to feline predators [11,12]. It is also suggested that exposure to *T. gondii* causes significant brain and behavioral anomalies in humans [13].

Molecular mimicry is a phenomenon where the antigenic determinants of microorganisms resemble those of the host, potentially triggering autoimmune responses. This theory has been extensively studied in the context of autoimmune diseases [14]. In mental disorders, this association is less clear. However, similarities between the intestinal microbiota and host antigens have been proposed to play a role in neurodegenerative diseases, including Alzheimer’s disease, Parkinson’s disease, and multiple sclerosis [14,15].

In silico analysis has facilitated the exploration of molecular mimicry in systemic diseases. Munera et al. demonstrated molecular mimicry between autoantigens associated with systemic lupus erythematosus and Leishmania species [16]. The similarity between human proteinase 3 and bacterial antigens has also been implicated in the pathogenesis of c-Antineutrophil cytoplasmic antibodies (ANCA)-associated vasculitis [17]. Furthermore, epitope prediction has been performed using bioinformatics tools, providing insights into cross-reactivity mechanisms [16,17]. This study aims to investigate molecular mimicry and cross-reactivity between *T. gondii* and human antigens related to schizophrenia, potentially elucidating their role in the development of schizophrenia associated with this infection.

## 2. Results

A total of 30 autoantigens implicated in schizophrenia were retrieved and analyzed: 10 from the AAgAtlas database and 20 from the literature (Appendix A). Only five autoantigens matched binary alignments using the criteria for having at least 30% identity with *T. gondii* proteins: *ENO2* (γ-enolase, UniProt: 09104), *TPO* (Thyroid peroxidase, UniProt: P07202), *GAD65* (Glutamic acid decarboxylase, UniProt: Q05329), *VRK2* (Serine/threonine-protein kinase *VRK2*, UniProt: Q86Y07), and *DPYD* (Dihydropyrimidine dehydrogenase, UniProt: Q12882) (Table 1).

The sequence identity between the autoantigens and proteins of *T. gondii* exhibited a wide range, reaching identities of up to 65%. The binary alignment for *ENO2* with *T. gondii* Enolase1 (GenBank: XP_002365578.1, strain ME49) and with *T. gondii* phosphoglycerate mutase 2 (PGAM2, UniProt: Q9BPL7), which showed the highest similarity, is presented (Figure 1). Both alignments revealed multiple conserved regions between *ENO2* and the parasite proteins: residues (40 to 58, 63 to 83, 91 to 98, 107 to 110, 129 to 133, 141 to 143, 145 to 150, 152 to 155, 157 to 162, 166 to 170, 175 to 178, and 183 to 188 for Enolase 1 (Figure 1a), and residues 09 to 25, 32 to 52, 59 to 68, 76 to 79, 98 to 102, 110 to 112, 114 to 119, 121 to 124, 126 to 131, 136 to 139, and 144 to 148 for PGAM2 (Figure 1b).

The autoantigen with the highest homology was *ENO2* (UniProt: P09104), which exhibited an identity of 65.39 and 64,03% with *T. gondii* enolases 1 (GenBank: XP_002365578.1, strain ME49) and Enolase 2 (GenBank: XP_002365579.1, strain ME49), respectively. The query cover of these alignments was over 98%, spanning almost the entire amino acid sequence. *TPO* (UniProt: P07202) showed homology with calcium-binding EGF domain-containing protein from several *T. gondii* strains, ranging from 32.08 to 47.83%, with the highest identity observed in the GT1 strain (EPR64714.1). However, the query coverage was low (4–15%). For *VRK2* (UniProt: Q86Y07), identities ranged from 30 to 34%, with query cover varying from 14 to 56%. The closest alignments were with *T. gondii* casein kinases from strains Me49 (GenBank: XP_018637608.1), VAND (KFH12280.1), GAB2-2007 (KFG46714.1), MAS (KFH16057.1), COUG (PIM04639.1) and GT1 (EPR63693.1). Partial identity was found for *GAD65* (UniProt: Q05329) and *DPYD* (UniProt: Q12882) against the *T. gondii* proteome, and epitope-level similarities were predicted, suggesting potential cross-reactivity.

All the human autoantigens were reported in the UniProt database. Models showed a typical fold expected for their protein family. The comparison of amino acid sequences revealed a high degree of identity between human *ENO2* and *T. gondii* enolases. Overlapping structural models of *ENO2* with *T. gondii* Enolase 1 demonstrated strong surface conservation, which may facilitate antibody recognition (Figure 2). Epitope modeling confirmed these conserved antigenic regions, which were visualized as antigenic patches (Figure 3). The RMSD value between human *ENO2* and *T. gondii* Enolase 1 was 0.759 Å, indicating high structural conservation. In contrast, the remaining four autoantigens did not show significant structural conservation with *T. gondii* proteins, although their models allowed identification of potential epitopes within antigenic patches.

Using the Ellipro server, linear and discontinuous epitopes were predicted for the schizophrenia-associated autoantigens, applying a threshold score > 0.7. For human enolase 2 (*ENO2*, UniProt ID: P09104), three linear epitopes were identified, which shared 66–68% sequence identity with *T. gondii* enolase 1 (GenBank: XP_002365578.1, strain ME49 (Table 2). To assess potential cross-reactivity, the predicted epitopes of *T. gondii* enolase were also modeled, and comparison of the antigenic patches revealed overlapping putative epitopes between human and parasite enolases (Figure 3).

For thyroid peroxidase (*TPO*, UniProt ID: P07202), four linear epitopes were retrieved, showing partial identity with the calcium-binding EGF domain-containing protein in different *T. gondii* strains: RUB (27%), FOU (29%), VEG (23%), and TgCatPRC2 (29%).

Vaccinia-related kinase 2 (*VRK2*, UniProt ID: Q86Y07) presented three linear epitopes with 25–28% identity to casein kinases from *T. gondii* strains CAST, VEG, and COUG (Table 2). Although no full-length sequence homology was detected for glutamate decarboxylase 65 (*GAD65*, UniProt ID: Q05329) or dihydropyrimidine dehydrogenase (*DPYD*, UniProt ID: Q12882) against the *T. gondii* proteome, epitope-level similarities were identified: *GAD65* antigenic patches showed 28% identity with a hypothetical helicase and 37% with a protein kinase, while no significant matches were found for *DPYD*.

To clarify the distribution of the *T. gondii* homolog proteins identified in this study across different genotypes, we constructed a Appendix A. This table summarizes the strains in which each homolog was detected, together with their accession IDs and genotype classification. The homologs were found both in clonal lineages (Type II: ME49, Type III: VEG) and in atypical or recombinant strains (e.g., CAST, RUB, FOU, TgCatPRC2, COUG). This diversity suggests that the potential molecular mimicry is not restricted to a single lineage of *T. gondii*, but rather may be conserved across distinct genotypes.

ConSurf analysis indicated that these autoantigen protein families exhibit variable degrees of structural conservation on the protein surface, supporting the possibility of partial cross reactivity (Figure 4).

## 3. Discussion

Immune dysregulation and autoantibodies have been linked to schizophrenia, suggesting the potential role of the immune response in its etiology [22,23]. This study investigated the potential cross-reactivity between autoantigens implicated in schizophrenia and proteins from *T. gondii*, an intracellular parasite previously associated with the disorder [5,24,25]. Through autoantigen analysis, homology-based modeling, and epitope prediction methods, we identified several autoantigens with similarities to *T. gondii* proteins, indicating a possible mechanism of molecular mimicry in the pathogenesis of schizophrenia.

Specifically, we identified significant similarities between five autoantigens implicated in schizophrenia and proteins from *T. gondii*. These autoantigens include gamma-enolase (*ENO2*), thyroid peroxidase (*TPO*), glutamic acid decarboxylase 65-kilodalton isoform (*GAD65*), serine/threonine-protein kinase 2 (*VRK2*), and dihydropyrimidine dehydrogenase [NADP(+)] (*DPYD*).

Notably, enolase exhibited the highest homology with *T. gondii* proteins, with identities reaching up to 65%. This finding aligns with previous studies suggesting that enolase, a glycolytic enzyme expressed ubiquitously in human tissues, is involved in both neurodegeneration and autoimmunity [26,27]. Some studies have found altered levels of enolase in the brains of individuals with schizophrenia [18,28]. Specifically, decreased levels of enolase have been observed in certain brain regions, suggesting a potential link between enolase dysregulation and schizophrenia pathology. Additionally, enolase has been implicated in processes related to neurodevelopment, neurotransmitter regulation, and neuronal survival, all of which are relevant to schizophrenia [27]. Identifying autoantigens sharing similarities with *T. gondii* proteins suggests a molecular mimicry mechanism in which the immune response against the parasite may inadvertently target self-antigens. This could contribute to disease development through the formation of autoantibodies, driven by conserved protein structures and amino acid sequences between *T gondii* proteins and human autoantigens related to schizophrenia.

Despite the lower sequence identity observed for other autoantigens, such as *GAD65* and *DPYD* (28–37%), these proteins still present regions of antigenic similarity that could elicit cross-reactive immune responses. Studies have demonstrated that even limited sequence homology in key immunogenic epitopes may suffice to trigger molecular mimicry, particularly if the antigenic determinants share a structural or conformational resemblance [29,30]. *GAD65*, for instance, is a well-known autoantigen in autoimmune neurological disorders [31], and its interaction with immune components could provide insights into the autoimmune processes underlying schizophrenia. Similarly, *DPYD*, an enzyme involved in pyrimidine metabolism, has been associated with neurological phenotypes [32], suggesting that even modest antigenic similarities could contribute to immune-mediated mechanisms in schizophrenia.

While bioinformatics has been employed to explore the connection between mental disorders and immunity [33,34], this study applies in silico analysis to schizophrenia autoantigens in a novel way. Our findings provide new insights into the potential mechanisms underlying mental disorder development. These results are significant as they reinforce the role of infections and the immune system in schizophrenia [35].

Despite the valuable insights from this research, several limitations should be acknowledged. The study relied on bioinformatics and computational modeling, which are constrained by the availability and accuracy of data inputs. In silico analysis has been helpful in correlating infections with the development of autoimmune responses [36,37,38]. In this study, validated in silico tools/databases and servers were applied, identifying potential autoantigens with identity and homology to antigens of *T. gondii*. This represents the first level of evidence in the search for mechanisms that could explain the development of schizophrenia in some patients. However, in vitro and in vivo studies are needed to investigate this hypothesis further, as in previous studies on other diseases [39].

It is worth noting that enolases are highly conserved and constitutively expressed across all *T. gondii* genotypes [40]. In contrast, other proteins such as casein kinases or calcium-binding EGF domain-containing proteins may show strain-specific divergence. Therefore, the potential for molecular mimicry could vary depending on the infecting genotype, warranting further studies including diverse clonal and atypical strains [41,42].

The results presented here are a preliminary approximation and justify experimental validation. Developing experiments such as ELISA to demonstrate cross-reactivity between schizophrenia-associated autoantigens and *T. gondii* antigens is essential, as these results could support future studies on the clinical impact of this phenomenon.

Additionally, the complexity of the immune system and its interaction with environmental factors in schizophrenia presents challenges in capturing all relevant variables [23]. The current study identified potential associations between autoantigens and *T. gondii* proteins but did not investigate causal relationships or mechanisms. Nevertheless, several immunopathogenic processes could underlie this interaction. Molecular mimicry may activate B and T cells against shared epitopes, promoting the production of cross-reactive autoantibodies [22,37]. Additionally, epitope spreading can occur, in which autoimmune recognition of self-antigens continues even after the pathogen has been cleared [43]. This phenomenon could be further amplified in the case of *T. gondii*, due to the persistence of the infection [5]. In parallel, alterations in blood–brain barrier integrity during infection can facilitate antibody and immune cell entry into the CNS, exacerbating neuronal damage [5,13]. Finally, chronic low-grade neuroinflammation, a phenomenon already described in schizophrenia, may act as a permissive context that amplifies these autoimmune responses [6,7]. Future studies with experimental validation and longitudinal analyses are needed to confirm the findings and elucidate the mechanisms involved.

In summary, these results suggest that the interaction between the immune system and environmental factors, such as *T. gondii* infection, could play an essential role in the pathophysiology of schizophrenia, with enolase emerging as a potential link factor. Furthermore, the possibility that *T. gondii* infection may contribute to schizophrenia development through immune system activation and the generation of autoantibodies targeting self-antigens underscores the complex interplay between environmental factors and immune dysregulation in the disease.

Finally, these findings highlight the value of in silico assays in clinical research on mental disorders. By leveraging computational methods, research hypotheses can be explored efficiently without exposing patients to risks. In silico analysis also facilitates the integration of multiple disciplines, fostering a holistic approach to studying and understanding complex diseases like schizophrenia.

## 4. Materials and Methods

### 4.1. Autoantigen Analysis

Two approaches were used to search for autoantigens for schizophrenia. First, a search was performed using the keywords “autoantigen” and “schizophrenia” in the Autoantigen database AAgAtlas, Beijing Proteome Research Center, Beijing, China. (http://biokb.ncpsb.org.cn/aagatlas_portal/index.php (accessed on 14 October 2025)) [44]. Second, using the keywords “autoantigen,” “schizophrenia,” and “autoimmunity” in PubMed, U.S. Department of Health and Human Services (HHS), Bethesda, MD, USA. (https://pubmed.ncbi.nlm.nih.gov/ (accessed on 14 October 2025)), a broader literature search was conducted [20]. Once autoantigens were identified using either approach, Research Collaboratory for Structural Bioinformatics Protein Data Bank RCSB-PDB, Rutgers University, Newark, NJ, USA (https://www.rcsb.org/ (accessed on 14 October 2025)) was searched to obtain the amino acid sequence of each autoantigen. These amino acid sequences were input in the Position-Specific Iterated **-** Basic Local Alignment Search Tool PSI-BLAST, National Center for Biotechnology Information, Bethesda, MD, USA. (https://blast.ncbi.nlm.nih.gov/Blast.cgi (accessed on 14 October 2025)) to find similar antigens in *T. gondii*. The search was limited to taxid 5811. Alignments with at least 30% sequence identity were considered relevant for subsequent analyses, since this threshold is generally accepted as indicative of biologically meaningful similarity in homology studies [45]. With the antigens found from the PSI-BLAST, a PRALINE PRofile ALIgNEment, Centre for Integrative Bioinformatics, Vrije Universiteit, Amsterdam, The Netherlands. (https://www.ibi.vu.nl/programs/pralinewww/ (accessed on 14 October 2025)) binary sequence alignment (set to BLOSUM62, with a gap penalty set to 12 in open and 1 in extension) was completed, comparing each antigen of *T. gondii* to the human autoantigen it was associated with. The *T. gondii* proteome sequences were retrieved from the National Center for Biotechnology Information NCBI Bethesda, MD, USA. (https://www.ncbi.nlm.nih.gov/) (last accessed 30 January 2024), including reference sequences from clonal lineages (ME49, VEG, GT1) as well as atypical or recombinant strains (CAST, RUB, FOU, TgCatPRC2, COUG, MAS, GAB2-2007).

### 4.2. Modeling Based on Homology

Three-dimensional (3D) structures from the autoantigens *VRK2* and *DPYD* were retrieved from Protein Data Bank (https://www.rcsb.org/ (accessed on 14 October 2025)) [46]. For autoantigens *ENO2*, *TPO*, and *GAD65*, 3D models were obtained using the Swiss Model server, Swiss Institute of Bioinformatics, Basel, Switzerland. (https://swissmodel.expasy.org/ (accessed on 14 October 2025)); those models were refined with Deep View for energy minimization. All templates used to generate theoretical models are reported in Table 1. The quality was evaluated by several tools, including the Ramachandran graphs, WHATIF, QMEAN4 index, and energy values (GROMOS96 force field). All models were visualized with Pymol 2.3, Schrödinger, New York, NY, USA. (https://www.pymol.org/ (accessed on 14 October 2025)) [47]. Structural homology was determined using the Root Mean Square Deviation index (RMSD) with Chimera, Resource for Biocomputing, Visualization, and Informatics (RBVI) University of California, San Francisco, CA, USA. (https://www.cgl.ucsf.edu/chimera/index.html (accessed on 14 October 2025)) [48]. To identify evolutionarily conserved regions potentially relevant for immune recognition, all 3D modeled structures were submitted to ConSurf-DB, Department of Biochemistry and Molecular Biology, George S. Wise Faculty of Life Sciences, Tel Aviv University, Tel Aviv, Israel. (https://www.ncbi.nlm.nih.gov/pmc/articles/PMC6933843/ (accessed on 14 October 2025)). These conserved residues on the protein surface were used to guide the prediction of cross-reactive epitopes. All parameters in ConSurf were set to default.

### 4.3. Epitope Prediction

B cell epitope prediction was performed with the Ellipro tool, The Immune Epitope Database (IEDB), San Diego Supercomputer Center, La Jolla, CA, USA. (http://tools.iedb.org/ellipro/ (accessed on 14 October 2025)) [49]. Prediction parameters were set up as default. Each epitope with a score > 0.70 was selected. Amino acid sequences of each autoantigen were used as input in PSI-BLAST (https://blast.ncbi.nlm.nih.gov/Blast.cgi (accessed on 14 October 2025)) to find similar antigens to *T. gondii* [50]. The protein with the highest max score was used in a PRALINE binary sequence alignment (https://www.ibi.vu.nl/programs/pralinewww/ (accessed on 14 October 2025)), comparing it to the autoantigens found in the first analysis to explore molecular mimicry between antigens from *T. gondii* and human autoantigens related to schizophrenia.

## Figures and Tables

**Figure 1 ijms-26-10321-f001:**
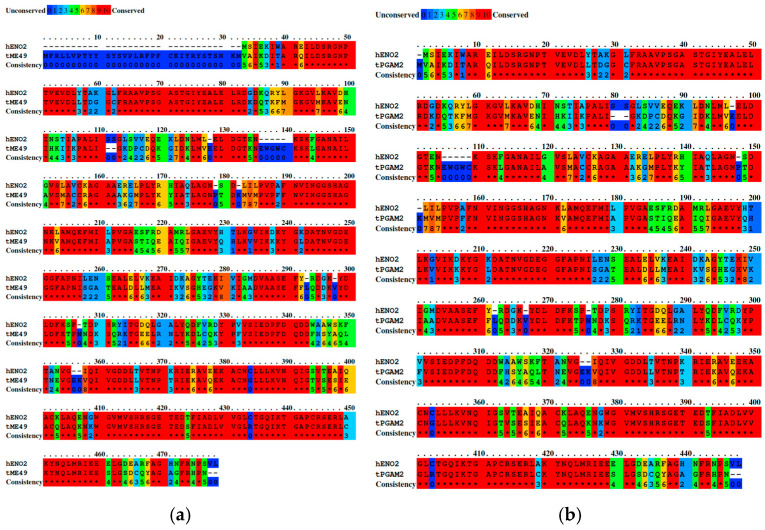
Binary alignment of human *ENO2* with its closest *T. gondii* homologs. (**a**) Alignment with *T. gondii* Enolase 1 (GenBank: XP_002365578.1, strain ME49), showing multiple conserved regions. (**b**) Alignment with *T. gondii* phosphoglycerate mutase 2 (PGAM2, UniProt: Q9BPL7), also reveals extensive conservation across several residues. Asterisks represent the same amino acid at the positions compared between the two sequences, the numbers indicate the positions of the amino acid residues relative to the total length of the protein.

**Figure 2 ijms-26-10321-f002:**
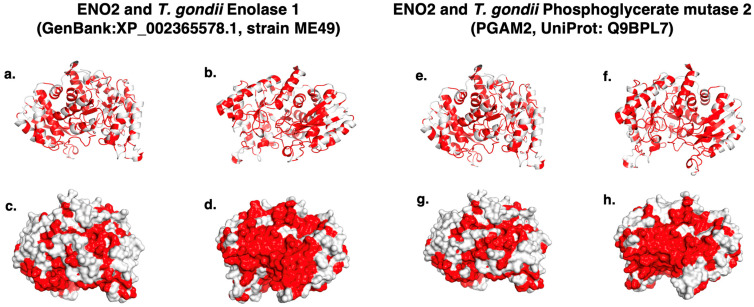
Surface and cartoon models of the overlapping of human *ENO2* with its closest *T. gondii* homologs. (**a**,**b**) Cartoon models, and (**c**,**d**) surface, of the alignment between human *ENO2* and *T. gondii* Enolase 1 (GenBank: XP_002365578.1, strain ME49), with conserved regions highlighted in red.(**e**,**f**) Cartoon models, and (**g**,**h**) surface models, of the alignment between human *ENO2* and *T. gondii* phosphoglycerate mutase 2 (PGAM2, UniProt: Q9BPL7), also showing conserved regions in red.

**Figure 3 ijms-26-10321-f003:**
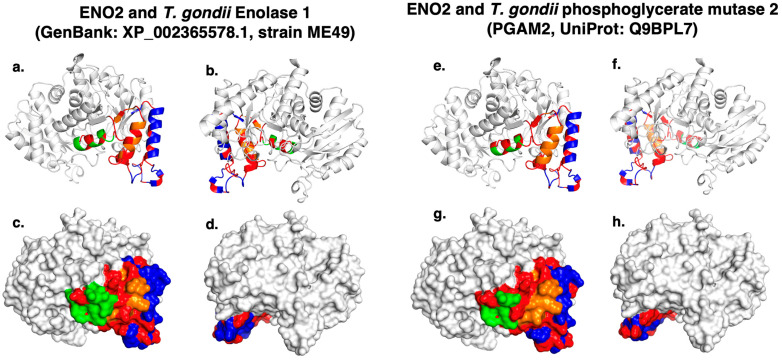
Surface and cartoon models of predicted epitopes of the overlap between human *ENO2* and its closest *T. gondii* homologs. (**a**,**b**) Cartoon models, and (**c**,**d**) surface models, of the alignment between human *ENO2* and *T. gondii* Enolase 1 (GenBank: XP_002365578.1, strain ME49), showing the position and surfaces covered by the predicted epitopes. (**e**,**f**) Cartoon models, and (**g**,**h**) surface models, of the alignment between human *ENO2* and *T. gondii* phosphoglycerate mutase 2 (PGAM2, UniProt: Q9BPL7), with predicted epitopes mapped on the surface. On surface patterns, all epitopes are indicated in color: blue (epitope 1), green (epitope 2), and orange (epitope 3). Conserved regions are shown in red.

**Figure 4 ijms-26-10321-f004:**
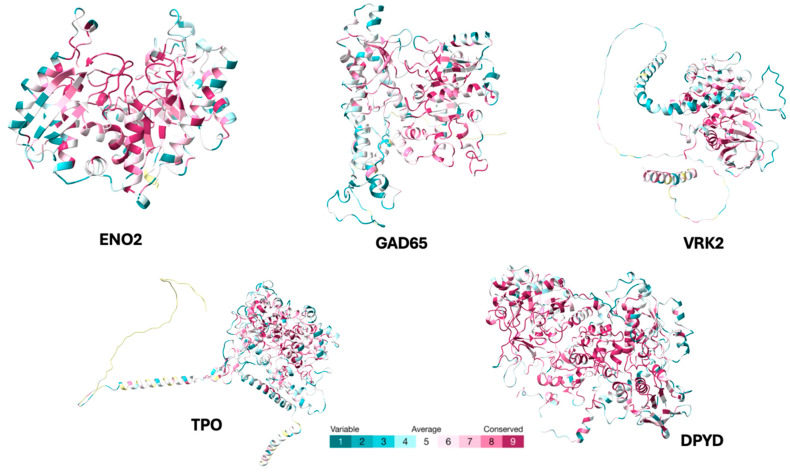
Surface models predicted by ConSurf. Purple indicates the highest conservation value among all autoantigens used in the study. On the right side, the purple region overlaps with the location of the antigenic patch, indicating that this area is highly conserved among the autoantigen protein family and could be involved in potential cross-reactivity.

**Table 1 ijms-26-10321-t001:** Autoantigens related to schizophrenia with a homology > 30% to antigens from *T. gondii*.

Autoantigen	Accession	Amino Acids	References
*ENO2*	P09104	434	Liu et al. [18]
*TPO*	P07202	933	Steiner J et al. [19]
*GAD65*	Q05329	585	Hansen N et al. [20]
*VRK2*	Q86Y07	324	Whelan R et al. [21]
*DPYD*	Q12882	1025	Whelan R et al. [21]

**Table 2 ijms-26-10321-t002:** Autoantigens with homology to antigens from *Toxoplasma gondii* and predicted cross-reactivity epitopes.

Autoantigen	Nº Epitope	Epitope	*T. gondii* Protein (Strain)	Accession	Identity (%)
*ENO2*	1	ASEFYRDGKYDLDFKSPTDPSRYITGDQLGA-LYQDFVRDYPVV	Phosphoglycerate mutase 2 (PGAM2, strain ME49)	Q9BPL7	68
	2	TNPKRIERA-VEEKACN	Enolase 1 (strain ME49)	XP_002365578	68
	3	DPFDQDD-WAAW-SKFTANVGIQI	Enolase 2 (strain ME49)	XP_002365579	66
*TPO*	1	ESVTDHVN-LITPLEKPLQN	Calcium-binding EGF domain-containing protein (strain FOU)	KFG44949	29
	2	TPLPA-NILDWQALN-YEIRGYVII	Calcium-binding EGF domain-containing protein (strain TgCatPRC2)	KYK67774	29
	3	LRQGYFVEAQPKIV	Calcium-binding EGF domain-containing protein (strain RUB)	KFG63764	27
	4	QEQDSYGGKFDR	Calcium-binding EGF domain-containing protein (strain VEG)	CEL74380	23
*GAD65*	1	RFKMFPEV-KEKGMAALPR-LI	Protein kinase (incomplete catalytic triad, strain ME49)	XP_018637623.1	37
	2	WYIPPSLRTLEERMSRLSKVA-PVIKAR	Putative helicase (partial, strain ME49)	PUA92631.1	28
*VRK2*	1	YCPNGNHKQYQENPRKGHN	Casein kinase (strain CAST)	RQX72942	28
	2	NPHGI-PLGPLDFSTKGQ	Casein kinase (strain COUG)	PIM01748	27
	3	YLAFPTNKPEKDAR	Casein kinase (strain VEG)	CEL78736	25

## Data Availability

The information in the databases used in this article is freely accessible and available for research purposes.

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
