# Peer review of "Molecular Mimicry Between *Toxoplasma gondii* B-Cell Epitopes and Human Antigens Related to Schizophrenia: An In Silico Approach"

_ijms, 2025, doi:10.3390/ijms262110321_

Round 1

Reviewer 1 Report

Comments and Suggestions for Authors

Using sequence analysis and epitope prediction, the authors identified five autoantigens (ENO2, TPO, GAD65, VRK2, and DPYD) that share similarities with Toxoplasma gondii proteins, with enolase showing the highest homology (up to 65%). These findings highlight shared antigenic features between T. gondii proteins and human antigens related to schizophrenia, supporting the potential role of T. gondii infection in schizophrenia pathogenesis. 

  1. The authors acknowledge limitations, but the bioinformatic results alone are not very sufficient to support the current conclusions drawn in the manuscript. After conducting the bioinformatic analysis, there were just few candidate genes. Are there any generated antibodies could be used to show the cross identification?
  2. Use consistent protein/gene names throughout the manuscript. For example, clarify whether “alpha-enolase”, “ENO2”, and “enolase 1” refer to the same protein. (Note: ENO1 and ENO2 are distinct genes in humans.)
  3. Provide the gene/protein IDs (from a single, consistent database such as NCBI Gene or UniProt) when each protein is first introduced. This prevents confusion from mixed naming conventions and makes it easy for readers to reproduce the searches.
  4. There is an apparent contradiction: the manuscript highlights GAD65 and DPYD among the five autoantigens, while Line 111 states that no similarity was found for GAD65 and DPYD when compared with the T. gondii proteome.
  5. Specify which T. gondii strains and proteome versions were searched (strain names and database versions/accession numbers).
  6. Line 103: “T. gondii enolases 1 y 2” does not seem right.
  7. Figure 1: Subpanels (a) and (b) are currently left/right and the details are hard to read. Consider arranging them top/bottom, enlarging the panels, increasing font sizes, and showing longer alignment windows.

Author Response

Response to Reviewers

We sincerely thank the reviewers for their thoughtful and constructive comments, which have helped us improve the quality and clarity of our manuscript. We have carefully considered each point raised and made the corresponding changes. Below, we provide a detailed, point-by-point response. All modifications are clearly indicated in the revised manuscript.

Review 1

Using sequence analysis and epitope prediction, the authors identified five autoantigens (ENO2, TPO, GAD65, VRK2, and DPYD) that share similarities with Toxoplasma gondii proteins, with enolase showing the highest homology (up to 65%). These findings highlight shared antigenic features between T. gondii proteins and human antigens related to schizophrenia, supporting the potential role of T. gondii infection in schizophrenia pathogenesis. 

  1. The authors acknowledge limitations, but the bioinformatic results alone are not very sufficient to support the current conclusions drawn in the manuscript. After conducting the bioinformatic analysis, there were just few candidate genes. Are there any generated antibodies could be used to show the cross identification?

Answer: We appreciate the comment. We recognize this limitation and have reinforced the Limitations and Discussion sections, indicating that this work is an initial in silico approach. For most identified candidate proteins, commercially available recombinant proteins and ELISA kits are accessible. Building on our previous experience with thyroperoxidase (TPO) as both an autoantigen and autoallergen in the context of chronic urticaria, the next planned step involves the development of ELISA inhibition assays. These assays, however, correspond to subsequent phases of the study that are currently under planning, as they require careful consideration of subject selection, appropriate sample types, and adherence to the corresponding ethical procedures.  (Page 1, lines 42—44; Page 10, lines 244-247; Page 11, lines 261-262)

  1. Use consistent protein/gene names throughout the manuscript. For example, clarify whether “alpha-enolase”, “ENO2”, and “enolase 1” refer to the same protein. (Note: ENO1 and ENO2 are distinct genes in humans.)

Answer: Thank you for noticing this. We have revised and corrected inconsistencies. We clarified from the first mention that ENO2 corresponds to gamma-enolase/neuron-specific enolase, distinct from ENO1. (Page 3, line 86)

  1. Provide the gene/protein IDs (from a single, consistent database such as NCBI Gene or UniProt) when each protein is first introduced. This prevents confusion from mixed naming conventions and makes it easy for readers to reproduce the searches.

Answer: We have added UniProt IDs for all proteins and antigens analyzed. (from page 3, lines 86—89 throughout the results section)

  1. There is an apparent contradiction: the manuscript highlights GAD65 and DPYD among the five autoantigens, while Line 111 states that no similarity was found for GAD65 and DPYD when compared with the T. gondii proteome.

Answer: Thank you for pointing this out. This was a writing error. The correct statement is: Partial identity was established for GAD65 (UniProt: Q05329) and DPYD (UniProt: Q12882) against the T. gondii proteome; epitope-level similarities were predicted, suggesting potential cross-reactivity. (Page 6, lines 120—122)

  1. Specify which T. gondii strains and proteome versions were searched (strain names and database versions/accession numbers).

Answer: Thank you for your suggestion. We have created supplementary Table 2, describing this information.

(Supplementary Material, page 13, table S2)

  1. Line 103: “T. gondii enolases 1 y 2” does not seem right.

Answer: This was corrected. Following your previous suggestion, we describe the proteins with their corresponding sources and IDs. (Page 5, lines 110—111)

  1. Figure 1: Subpanels (a) and (b) are currently left/right and the details are hard to read. Consider arranging them top/bottom, enlarging the panels, increasing font sizes, and showing longer alignment windows.

Answer: We appreciate your comment. We have redesigned the figure vertically, with higher resolution and a larger font size.

Reviewer 2 Report

Comments and Suggestions for Authors

This is a very interesting paper dealing with an intriguing features of Toxoplasma gondii Infection and suggest a possible  correlation between toxoplasmosis and schizophrenia  due to the mimicring of toxoplasma antigens and human autoantigens.all the study was well conducted and conclusions very clear. The main limitation that is that the study was relied on bioinformatics and computational modeling and did not investigate the underlying mechanisms.   It could represent a relevant starting point to  investigate this hypotesis with in vivo study

Author Response

Response to Reviewers

We sincerely thank the reviewers for their thoughtful and constructive comments, which have helped us improve the quality and clarity of our manuscript. We have carefully considered each point raised and made the corresponding changes. Below, we provide a detailed, point-by-point response. All modifications are clearly indicated in the revised manuscript.

Review 2

This is a very interesting paper dealing with an intriguing features of Toxoplasma gondii Infection and suggest a possible  correlation between toxoplasmosis and schizophrenia  due to the mimicring of toxoplasma antigens and human autoantigens.all the study was well conducted and conclusions very clear. The main limitation that is that the study was relied on bioinformatics and computational modeling and did not investigate the underlying mechanisms.   It could represent a relevant starting point to  investigate this hypotesis with in vivo study

Answer: We appreciate the observation. We strengthened the Limitations section to acknowledge this point and proposed experimental studies as the next step to confirm the findings.  (Page 1, lines 42—44; Page 10, lines 244-247; Page 11, lines 261-262)

Reviewer 3 Report

Comments and Suggestions for Authors

The manuscript addresses a highly relevant topic, as the potential relationship between Toxoplasma gondii infection and the development of schizophrenia represents an interdisciplinary field of research with both biomedical and public health implications.

Review the formatting of scientific names throughout the text and in the supplementary material, as all of them must be written in italics.

You mentioned that: line 84-85 “Only five autoantigens matched binary alignments using the criteria for having at least 30% identity with T. gondii proteins”; Is there a minimum percentage of similarity that autoantigens must present to be associated with schizophrenia? If there is a minimum percentage, it should be included.

Are the Toxoplasma gondii proteins studied in this work constitutively expressed across all clonal, recombinant, and atypical genotypes?

The manuscript would be enriched by including a table specifying the protein and the genotype in which it is present, whether clonal, atypical, or recombinant.

Line 224-225. “The current study identified potential associations between autoantigens and T. gondii proteins but did not investigate causal relationships or mechanisms”. It would be interesting to mention some possible mechanisms.

Author Response

Response to Reviewers

We sincerely thank the reviewers for their thoughtful and constructive comments, which have helped us improve the quality and clarity of our manuscript. We have carefully considered each point raised and made the corresponding changes. Below, we provide a detailed, point-by-point response. All modifications are clearly indicated in the revised manuscript.

Review 3

The manuscript addresses a highly relevant topic, as the potential relationship between Toxoplasma gondii infection and the development of schizophrenia represents an interdisciplinary field of research with both biomedical and public health implications.

-Review the formatting of scientific names throughout the text and in the supplementary material, as all of them must be written in italics.

Answer: We reviewed the full text and corrected the nomenclature of Toxoplasma gondii and other species.

-You mentioned that: line 84-85 “Only five autoantigens matched binary alignments using the criteria for having at least 30% identity with T. gondii proteins”; Is there a minimum percentage of similarity that autoantigens must present to be associated with schizophrenia? If there is a minimum percentage, it should be included.

Answer: We thank the reviewer for this observation. We have now added a justification for the ≥30% sequence identity threshold in the Methods section, supported by the bioinformatics literature. Specifically, we cite Pearson (2013), who notes that alignments above this cutoff are generally considered biologically meaningful in homology studies. (Page 11, lines 285—287)

-Are the Toxoplasma gondii proteins studied in this work constitutively expressed across all clonal, recombinant, and atypical genotypes?

Answer: Thank you for pointing this out. We added a note in the Discussion about variability in expression among clonal, recombinant, or atypical genotypes, noting which are constitutive. (Page 10, lines 239—243)

-The manuscript would be enriched by including a table specifying the protein and the genotype in which it is present, whether clonal, atypical, or recombinant.

Answer: We appreciate your suggestion. We created a supplementary table (Supplementary Table 2) indicating the strains/genotypes where homologs were identified. (Supplementary Material, page 13, table S2)

-Line 224-225. “The current study identified potential associations between autoantigens and T. gondii proteins but did not investigate causal relationships or mechanisms”. It would be interesting to mention some possible mechanisms.

Answer: Thank you for your comment. We expanded the Discussion, suggesting pathogenic mechanisms: B/T cell activation against shared epitopes, autoantibody production, blood-brain barrier alterations, and chronic inflammation. (Page 10, lines 251—262)